# Selenium Biofortification of Crop Food by Beneficial Microorganisms

**DOI:** 10.3390/jof6020059

**Published:** 2020-05-03

**Authors:** Yuanming Ye, Jingwang Qu, Yao Pu, Shen Rao, Feng Xu, Chu Wu

**Affiliations:** College of Horticulture & Gardening, Yangtze University, Jingzhou 434025, China; y.m.ye2020@gmail.com (Y.Y.); qujw@yangtzeu.edu.cn (J.Q.); puyao456@163.com (Y.P.); raoshen1989@163.com (S.R.); xufeng198@126.com (F.X.)

**Keywords:** selenium, biofortification, transporters, mycorrhizal fungi, plant growth-promoting rhizobacteria (PGPRs)

## Abstract

Selenium (Se) is essential for human health, however, Se is deficient in soil in many places all around the world, resulting in human diseases, such as notorious Keshan disease and Keshin–Beck disease. Therefore, Se biofortification is a popular approach to improve Se uptake and maintain human health. Beneficial microorganisms, including mycorrhizal and root endophytic fungi, dark septate fungi, and plant growth-promoting rhizobacteria (PGPRs), show multiple functions, especially increased plant nutrition uptake, growth and yield, and resistance to abiotic stresses. Such functions can be used for Se biofortification and increased growth and yield under drought and salt stress. The present review summarizes the use of mycorrhizal fungi and PGPRs in Se biofortification, aiming to improving their practical use.

## 1. Introduction

At present, it is widely accepted that selenium (Se) possesses multiple physiological functions in various biological systems as an integral part of a range of proteins containing Se. Therefore Se is important for human health. However, Se distribution in the earth’s crust is greatly uneven, ranging from 0.005 mg·kg^−1^ in Finland to 8000 mg·kg^−1^ in Tuva-Russia [1]. Se deficiency has been reported in many places all around the world including China, North America, New Zealand, Australia, Sweden, and Finland [2,3,4,5]. Some notorious diseases are directly related to Se deficiency, such as Keshan disease and Keshin–Beck disease, two endemic diseases related to Se deficiency. Keshan disease was first prevalent at alarge scale in 1935 in Keshan county, Heilongjiang province, China. Keshan disease generally occurs in children and women of childbearing age and its symptoms are related to impairment of cardiac function, cardiac enlargement, and arrhythmia [6]. Although the main factor was not determined for the disease in etiology, it was closely related to Se because it was found that there was an obvious Se deficiency in local soil, and Se supplementation could partly control the disease. An investigation analyzed some physiological parameters, including blood Se level, glutathione peroxidase-1 (GPx-1) activity, and variance at codon 198 in *GPx-1* gene, and found that the main risk factors for the disease were low GPx-1 activity, Keshan disease family history, and living in an endemic area [7], suggesting that Keshan disease is closely related to low GPx-1 activity. Kaschin–Beck disease is an osteoarthropathy, which manifests as severe dysarthrosis of joints, shortened fingers and toes, and in severe cases dwarfism. In China, the disease is prevalent in the Tibetan Plateau [8,9]. An investigation carried out by Zhang et al. [8] showed that the levels of environmental Se were very low, and Kaschin–Beck disease in the Tibetan Plateau was much severe with decreasing environmental Se under the Se-deficient condition, suggesting the relationship between Kaschin–Beck disease and Se deficiency in the Tibetan Plateau. In addition, Se is related to other human diseases and health, such as cancer [10,11,12,13], muscle disease [14], and healthy aging and longevity [15,16,17]. Therefore, it is essential to maintain Se homeostasis in human body [18,19]. It was estimated that Se intake of >900 μg·day^−1^ is harmful, and intake of <30 μg·day^−1^ is not enough [20]. Some data have shown that over 800million people all around the world might suffer from Se deficiency [21,22,23,24,25,26]. Therefore, sufficient dietary Se uptake is important for human health.

Acquired Se is converted into some proteins thatcontain at least one of the two amino acids (i.e., selenocysteine (SeCys) and selenomethionine (SeMet)) as a key component (i.e., selenoproteins). Human health and diseases are related to selenoproteins, and selenocysteine is regarded as the 21st proteinogenic amino acid. The human genome encodes about 30 selenoproteins. In the article written by Reeves and Hoffmann [27], they described functions of selenoproteins in detail. Among the selenoproteins in human, glutathione peroxidases (GPxs) seem to be more important, because they include eight proteins (GPx1–GPx8) having antioxidant properties with multidimensional roles in living cells, ranging from H_2_O_2_ homeostasis to regulation of apoptosis [28]. Therefore, enough Se uptake is essential for functional maintenance of these selenoproteins. Since Se is deficient in many places all around the world, Se fortification in food is necessary. In view of high toxicity of selenite and selenate, Se biofortification is relatively bio-safe. Organic Seleno-compounds act as potential therapeutic and chemo-preventive agents that function as antioxidants, enzyme modulators, antitumor, antimicrobials, antihypertensive agents, antivirals, and cytokine inducers [29]. Organic seleno-compounds are provided with crop food [30,31,32,33,34,35], vegetables [36,37,38,39,40,41,42], fruits [43,44,45], and even nuts [46,47,48]. Therefore, how to increase concentrations of organic seleno-compounds in these plants is of significance for improvement of dietary Se acquisition by human being. 

Se biofortification may be carried out by multiple ways, such as application of Se fertilizers on leaves [49,50,51,52,53] and in soil [52,54,55]. Se-enriched organic fertilizers are also applied. For example, Bañuelos et al. [56] used Se-enriched *Stanleya pinnata* to cultivate Se-enriched broccoli and carrots, and found that more than 90% of organic Se was converted to inorganic selenate and selenite. Se foliar application seems to be most effective way to fortify Se uptake in most arable crops [52,57]. However, a contrary result was observed by Lyons et al. [58]. They found foliar application was less efficient than application to soil at planting (at application rates of 40 and 120 g·ha^−1^, respectively) in Australian trials. The agronomic application of Se fertilizers are more expensive and short-term solutions, especially in large-scale fields. Relatively, agronomic Se biofortification with beneficial microorganisms (BMOs) is a more inexpensive and long-term solution, especially in poor places and Se-rich places, such as Enshi, Hubei province, China [59] and Pineridge Natural Area, a seleniferous site west of Fort Collins, CO, USA [60]. 

In the present article, we focus on the roles of beneficial microorganisms in Se biofortification and our aim is to improve use of beneficial microorganisms in practice.

## 2. Improvement of Se Biofortification by BMOs

Symbiosis of plants with BMOs is helpful for plant growth and to increase in micronutrition uptake and resistance to abiotic and biotic stresses. Based on the characteristics of BMOs, BMOs can be used for Se biofortification. BMOs, including mycorrhizal fungi (endo- and ectomycorrhizal fungi), root endophytic fungi (REFs), and PGPRs, are popular in biofilmed biofertilizers. Arbuscular mycorrhizal fungi (AMFs) are preferential to colonize in roots of angiosperms, and ectomycorrhizal fungi are popular in gymnosperms. Most REFs possess a wide range of plant hosts.

### 2.1. Arbuscular Mycorrhizal Fungi

Arbuscular mycorrhizal fungi are used for Se biofortification because of their ability to enhance nutrition uptake of their host plants (Table 1). Functions of mycorrhizal fungi have been the primary focus of research, especially those involved in phosphate uptake. The genomes of these fungi encode some high-affinity inorganic phosphate transporters and some of them have been isolated and identified [61,62,63,64,65]. On the other hand, *in planta*, some symbiosis-specific phosphate transporters can be induced by symbiosis with mycorrhizal fungi [61,66,67,68,69,70,71,72]. Thus, the interaction between plants and mycorrhizal fungi strengthens phosphate uptake and transportation to host plants [73,74,75,76,77]. Similarly, there are some sulfate transporters encoded by genomes of mycorrhizal fungi, such as sulfate transporters GBC38160.1 and GBC25943.1 and sulfate permeases PKY50973.1 in arbuscular mycorrhizal fungus *Rhizophagus irregularis*, sulfate transporters EDR02618.1 and EDR02177.1, and sulfate permeases EDR11271.1 and EDR00466.1 in ectomycorrhizal fungus *Laccaria bicolor*. Since Se and sulfur (S) belong to the same element family (VI-A), the chemical properties of Se are very similar to S. Se is absorbed as selenate or selenite, which is metabolized via the sulfur assimilation pathway in plants, leading to biosynthesis of SeCys, SeMet, and other Se isologs of various S metabolites [78,79,80,81,82]. Se can be transported by sulfur transporters to host plants, just like phosphate transported between mycorrhizal fungi and their host plants, such as the high-affinity sulfate permease [83] and the high-affinity sulfate transporters Sultr1:1 and Sultr1:2 [84,85,86]. The two sulfate transporters are proton-sulfate symporters, such that for every molecule of selenate entry into root cells, three protons are taken up. Sulfate transporters function in Se accumulation in food crops. Wheat genotype ‘Puelche’ is the most Se-tolerant and has the greatest Se accumulation among the three wheat genotypes studied (i.e., ‘Puelche’, ‘Tinto’, and ‘Kumpa’), such that its Se accumulation was related to the strongest transcript level of the sulfate transporter TaeSultr4.1 in roots [87]. In addition, other transporters also take part in Se transport, such as silicon transporters in rice [88] and tomato [89], phosphate/orthophosphate transporters in wheat [90], rice [91,92,93], tomato [89], and yeast (*Saccharomyces cerevisiae*) [94,95], and monocarboxylates transporters in yeast (*S. cerevisiae*) [96]. Thus, it is reasonable to explain the experimental results that plant availability of selenate and selenite was influenced by the competing ions phosphate and sulfate [97,98]. Competition between phosphate and Se uptake led to decrease in Se accumulation translocation coefficients, and Se concentrations in wheat roots, stems, leaves, and spikes when phosphate fertilizers were applied to selenite fertilized soil [99]. However, a different case occurred. An investigation was carried out on sulfate and selenate uptake in *Astragalus* species (two Se hyperaccumulators *A. racemosus* and *A. bisulacatus* and two closely related non-accumulators *A. glycyphyllos* and *A. drummondii*), and results showed that sulfur deficiency increased Se accumulation, and increased Se supply increased sulfate accumulation in both root and shoot tissues [100]. In certain *Astragalus* species, the high expression of sulfate transporters led to enhanced ability of Se uptake and translocation, and therefore contributed to the Se hyperaccumulation trait. At present, except for sulfate and phosphate transporters, it is not clear whether other transporters have their homologous proteins in mycorrhizal fungi. If these homologous proteins occur in mycorrhizal fungi, they could mediate Se transport to host plants. On the other hand, decreases in sulfate bioavailability and mycorrhizal symbiosis enhanced expression of sulfate transporters, resulting in increase in ability to absorb sulfate and consequent uptake of Se [101,102,103,104]. Similarly, Se deficiency also enhances expression of sulfate transporters, resulting in an increase in Se uptake, and mycorrhizal symbiosis also enhances Se uptake.

Some evidence supports the role of mycorrhizal fungi in enhancing Se uptake in plants. Wheat seedlings were inoculated with *Glomusversiform* or *Funneliformis mosseaein* hydroponic culture medium for eight weeks, the two arbuscular mycorrhizal fungi significantly increased selenate and selenite uptake by wheat root, but they did not show effect on uptake of SeMet [105]. Meanwhile, compared to non-mycorrhizal roots, mycorrhizal roots showed significantly higher *V*_max_ for selenate and selenite uptake (179.6 vs. 55.93 nmol·g^−1^DW·h^−1^ for selenate and 1688.0 vs. 860.3 nmol·g^−1^DW·h^−1^ for selenite). Higher Se accumulation was carried out through up-regulating the expression of three genes encoding sulfate transporters, i.e., TaSultr1:1, TaSultr1:3, and TaSultr2:1, in the mycorrhizal roots, especially TaSultr1:1. In mycorrhizal roots with G. *versiform* and *F. mosseae*, the relative expressions of *TaSultr1:1* gene was significantly up-regulated by 2.18-fold and 2.12-fold, respectively. Garlic (*Allium sativum* L.) is an important condimental species. This species ispopular all around the world because of its diallyl disulfide, a component of garlic, which can inhibit proliferation of various cancer cells (e.g., colon, lung, and skin cancer cells) and WEHI-3 leukemia cells [115,116,117]. Garlic is used for Se biofortification with mycorrhizal fungi. A survey of applying selenate fertilizer and mycorrhizal fungus *Glomus irtraradices* to soil was conducted, and the results showed mycorrhizal addition increased the Se uptake of garlic by10-fold to 15 μg·g^−1^DW, and fertilization with selenate and amendment of mycorrhizal fungi strongly increased the Se concentrations in garlic to around 1% [55]. Further analyses showed that the amendment of soil with the mycorrhizal fungus and/or selenate increased selenate concentrations in garlic, but did not affect distribution of detected Se species in garlic. In Se-contaminated soil, mycorrhizal fungi inoculation increased Se accumulation of plants. Alfalfa, maize, and soybean seedlings were cultivated in the soil contaminated with different levels of Se, and results showed that mycorrhizal fungi inoculation decreased Se accumulation in roots and shoots of all the plants at low Se levels (0 or 2 mg·kg^−1^), but increased Se accumulation in alfalfa shoots and maize roots and shoots at Se level of 20 mg·kg^−1^ [112]. Contrary results were observed on ryegrass (*Lolium perenne* cv. ‘Barclay’) [111]. Their results showed that Se concentrations in roots of ryegrass were not affected by mycorrhizal inoculation with the AMF *G. mosseae*, but mycorrhizal inoculation significantly reduced Se concentrations in shoots [111], further decreasing Se uptake in whole plants. Lettuce (*Lactuca sativa* L.) is one of the most consumed leaf vegetables in some places around the world because of its good properties, such as high levels of antioxidants (such as carotenoids, polyphenols, ascorbate, α-tocopherol) and dietary fiber [118,119,120], thus it is suitable for Se biofortification to enhance dietary Se consumption. When two lettuce cultivars ‘Batavia Rubia Munguia’ (BRM) and ‘Maravilla de Verano’ (MV) were treated with Se compounds (selenite, organic Se compounds SeU and SeCH_3_) and AMFs (a mixture of *Rhizophagus intraradices* and *Funneliformis mosseae*), their growths were continuously improved by AMFs, except for BRM under treatment of SeCH_3_ [106]. The positive effect of AMFs on plant biomass was different among lettuce cultivars and forms of seleno-compounds, and BRM lettuce plants showed the highest mycorrhizal efficiency index (MEI) under treatment of SeU, MV lettuce plants with the highest MEI under SeCH_3_, suggesting that the two lettuce cultivars possessed preference for different seleno-compounds when they were inoculated with AMFs. Meanwhile, AMFs inoculation significantly affected mineral accumulation in the leaves of BRM lettuce. In general, mycorrhizal inoculation significantly increased levels of macro and micronutrients, but significantly reduced Se levels in leaves of BRM lettuce. Significant interaction occurred about Se levels in shoots of BRM lettuce between seleno-compounds and AMFs inoculation. Similar status occurred on MV lettuce. Under treatment of selenite, AMFs inoculation reduced Se concentrations in leaves of MV lettuce. In contrast, under treatment of organic seleno-compound SeCH3, MV lettuce never accumulated detectable levels of Se in leaves, regardless of whether they were inoculated with AMFsor not. Treatment of organic seleno-compound SeU slightly increased Se concentrations in leaves of MV lettuce without AMFs inoculation [106]. Other research showed similar results [108]. All the results suggest that combination of seleno-compounds and AMFs inoculation does not increase Se levels in lettuce leaves, although it increases levels of some macro- and micronutrients and antioxidants. Therefore, some AMFs are not suitable for Se biofortification in lettuce. Of course, other AMFs should be chosen to examine their role in Se biofortification in lettuce under treatment of seleno-compounds. At present, it is not clear whether lettuce symbioses with some ectomycorrhizal fungi. Thus, more research is necessary for Se biofortification in lettuce.

Consversa et al. [107] investigated the effect of Se fern application and AMFs (*Rhizophagus intraradices* and *Funneliformis mosseae*) inoculation on Se biofortification for two years, such that Se fern application was carried out on green asparagus (*Asparagus officinalis* L.). Their experimental results showed that Se levels in non-mycorrhizal *A. officinaliscv*. ‘Grande’ plants increased in trial A1 as exogenous selenate levels increased. Under selenate treatment of 75 and 125 g·ha^−1^, Se concentrations in spears increased 4.7 and 6.4-fold on a dry weight basis compared to control, respectively. Similar results occurred in trail B1. In trail B1, Se concentrations in spears were significantly affected by the interaction between Se amendment and AMFs inoculation. In spears of plants without Se amendment, Se levels were similar in mycorrhizal and non-mycorrhizal plants. All the results suggest a combination of Se amendment and mycorrhizal fungi greatly improve Se biofortification in *A. officinalis* and the combination should be recommended in field by large scale. However, contrary results have also been observed. When the AMF *Glomus mosseae* was used for inoculation with alfalfa (*Medicago sativa* L. cv. ‘Chuangxin’), maize (*Zea mays* cv. ‘ND108′), and soybean (*Glycine max* cv. ‘Zhonghuang No. 17′), mycorrhizal inoculation significantly decreased Se concentrations in roots with the highest reduction for alfalfa (50–70%), while it was less than 40% for maize and soybean, Se concentrations in shoots decreased by 7–38% for mycorrhizal treatment, and the difference caused by inoculation influence was insignificant among the plant species [112]. When Se was added at the levels of 0 and 2 mg·kg^−1^, the total Se accumulation in roots and shoots of all the three plant species were lower in mycorrhizal than in non-mycorrhizal treatment, while the opposite pattern was observed in roots of maize and shoots of alfalfa and maize when Se was applied at 20 mg·kg^−1^ [112]. These results show negative effects on Se accumulation in these plant species when low levels of exogenous Se were added.

In addition, some ectomycorrhizal fungi can accumulate Se in their fruit bodies [121,122,123], suggesting their ability to acquire Se. Some of these ectomycorrhizal fungi are edible, thus they are used for biofortification of Se in fruit bodies. Few investigations on the role of REFs in Se biofortification have beencarried out (Table 1). In general, REFs, especially members of the genus *Trichoderma*, can colonize roots of some host plants, thus they can be widely used for Se biofortification. At present, there are not reports on roles of dark septate fungi in Se biofortification of food crops.

Taken together, mycorrhizal inoculation might increase Se accumulation in some crop species, leading to Se biofortification of crops. For some crop species, more investigations are needed, especially for interactions between mycorrhizal fungi and crop species. For the abovementioned negative effects of G. *mosseae* on Se accumulation in alfalfa, maize, and soybean, more mycorrhizal fungi and root endophytic fungi should be used to investigation.

### 2.2. Se Biofortification by PGPRs

Plant growth-promoting rhizobacteria (PGPRs) are popular in improving nutrition uptake, plant growth, and resistance to abiotic stresses [124,125,126,127]. Some of them possess the ability to solubilize phosphate in soil. Such ability could be used for Se biofortification (Table 2), because in some soil agrotypes, such as volcanic Andisols in southern Chile, Se bioavailability is very low. On the one hand, Se can form stable complexes with clays and/or can be strongly absorbed onto oxy-hydroxides of aluminum, iron, or manganese, and remain low in terms of bioavailability to plants [128,129,130]. On the other hand, oxyanions of Se, i.e., selenite and selenate, are bioavailable to plants. When selenate and selenite are supplied to soil, they are rapidly reduced to insoluble forms (e.g., Se–metal ion complex), leading to their low bioavailability (less than 10% only). The Se fertilizers that are not acquired by plant roots readily after application are not bioavailable to plants in the next season or the next year [131]. Thus, Se re-solubility in soil is very important. Although there are no report concernsregarding Se-solubilizing PGPRs at present, some seleno-bacteria have been studied [30,35,109,132,133]. Trivedi et al. [35] isolated and identified some endophytic seleno-bacteria from the various tissues of *Ricinus communis* plants and molecular identification analyses showed that they were *Paraburkholderia megapolitana*, *Alcaligenes faecalis*, and *Stenotrophomonas maltophilia*. Among the three bacteria, *P. megapolitana* was most effective in improving the growth of *Glycine max* plants under drought and enhancing Se biofortification which was 7.4-fold higher compared to control. The synergistic effect on Se biofortification and increased drought tolerance is important for plants grown in arid and semi-arid places with Se deficiency. A great number of people all around the world are dependent on wheat as their main component of diet, thus it is important to fortify Se in wheat grains [30]. Many studies have been carried out on Se biofortification in wheat. Durán et al. [109] evaluated the effects of Se acquisition by wheat plants through the co-inoculation of native seleno-bacteria strains *Stenotrophomonas* sp. B19, *Enterobacter* sp. B16, *Bacillus* sp. R12, and *Pseudomnas* sp. R8, both individually and in mixture, as a seleno-nanosphere source with AMF *Glomus claroideum*. They found that Se concentrations in plant tissues in inoculated plants were significantly higher than those of un-inoculated controls.Meantime, regardless of presence of AMF *G. claroideum*, Se concentrations in grains of wheat plants inoculated with *Enterobacter* sp. B16 were higher than those of plants inoculated with the rest of the microbial strains. In addition, PGPRs showed their synergistic role in improving Se concentrations with AMFs. When plants were inoculated with the seleno-bacteria strains and *G. claroideum*, Se concentrations in grains were 23.5% higher than those in non-mycorrhizal plants. The synergisms might be related to the relationship between seleno-bacteria strains and AMFs, because the seleno-bacteria could acquire more nutrition from the hyphae of their neighboring AMFs or ectomycorrhizal fungi [134,135,136,137]. Moreover, Durán et al. [132] isolated two Se-tolerant endophytic bacteria *Acinetobacters* sp. E6.2 and *Bacillus* sp. E5. They studied production of seleno-compounds (SeMet and seleno-methyl-selenocysteins (MeSeCys)) by the two bacteria, but they did not study the effects of the two bacteria on Se biofortification. Co-application of Se fertilizers and seleno-bacteria sometimes leads to changes in bacterial population. When Se-tolerant bacteria and Se amendment were supplied to wheat in Andisols, Se amendment stimulated population growth of two bacterial groups (*Paenibacillaceae* and *Brucellaceae*), but inhibited other bacterial groups (Clostridia, *Burkholderiales*, *Chitinophagaceae*, and *Oxalobacteraceae*) [133]. Meanwhile, Se concentrations in roots and leaves of wheat plants inoculated with Se-tolerant bacterial strains *Pseudomonas* sp. R8 and *Stenotrophomonas* sp. B19 were significantly higher than those of the un-inoculated controls. Higher Se biofortification is related to the Se tolerance of the two bacteria, because higher Se concentrations in roots and leaves were also observed when wheat plants inoculated with *Stenotrophomonas* sp. B19 were grown at concentrations of 5 and 10 mM of selenite, compared to those grown at 2 mM [133]. The results suggested that Se in seleno-bacteria could be transferred into their host plants. Effects of other Se-tolerant bacteria on Se biofortification were also investigated. When wheat plants were inoculated with two Se-tolerant bacterial strains *Bacillus cereus* YAP6 and *Bacillus licheniformis* YAP7, Se concentrations in the stems of the Se-treated wheat plants were increased up to 375%, and Se concentrations in kernels increased up to 154% of those in un-inoculated Se-treated wheat plants [34]. Meanwhile, the *Bacillus* strains can produce auxin, leading to increased number of leaves and greater biomass and shoot length [34]. When wheat plants were inoculated with *Bacillus pichinotyi* in the presence of selenate, they posed significantly higher biomass, shoot length, and spike length compared to un-inoculated plants [33]. Meanwhile wheat plants inoculated with *B. pichinotyi* had significantly higher Se concentrations in wheat kernels (167%) and stems (252%), compared to un-inoculated plants. Overall, greater biomass means higher Se biofortification, which is important for crops cultivated by large scale in field. Rhizobia not only fixes nitrogen, but also helps Se accumulation. Data from Alford et al. [138] showed rhizobia significantly increased shoot biomass and Se accumulation in shoots of the Se-hyperaccumulator *Astragalus bisulcatus* and the nonhyperaccumulator *A. drummondii*. The dual roles of rhizobia are of significance for organic Se production.

Interestingly, volatile organic compounds (VOCs) released by PGPRs improve Se biofortification of plants. VOCs from *Bacillus amyloliquefaciens* BF06 significantly increased photosynthesis and growth of *Arabidopsis* plants and these VOCs led to an obvious increase in expressions of some genes encoding sulfate transporters and Se concentrations in plants [141]. VOCs released by *B. amyloliquefaciens* could not increase Se biofortification of *Arabidopsis Sultr1:2* mutants. All the results suggested sulfate transporters with high expression mediate Se uptake, as shown above. Meanwhile, the results indicate an unknown mechanism that PGPRs improves Se biofortification. The question is inevitable, how do the VOCs improve expression of sulfate transporters?

Taken together, Se amendment could improve population growth of some Se-tolerant bacteria; if these bacteria show synergistic effect on Se biofortification, they could be mixed in some biofilmed biofertilizers specific to certain crops and vegetables, thus, their combinative amendment along with Se fertilizers become a Se biofortification tool in sustainable agriculture [52,145].

## 3. Concluding Remarksand Perspectives

Since Se is essential for human health, Se biofortification must be carried out in Se-deficient places by various ways on food crops, vegetables, fruits, and nuts. Foliar and soil fertilization are effective for enhancing Se accumulation in crops. However, the two ways are expensive for large-scaled food crops, especially in poor places. Moreover, the effect of the two ways is short-term and they easily cause area source pollution. BMOs improve Se uptake and accumulation in food crops. Therefore, combination of BMOs and soil fertilization is a good approach to Se biofortification of crop food. At present, for Se biofortification by BMOs, there remain some questions to resolve. The first relates to the synergisms among these beneficial microorganisms. Biofilmed biofertilizers often include many BMOs. As they can compete for nutrition from their common host plants, some of them are possibly antagonistic. Therefore, before they are mixed in biofilmed biofertilizers, the synergism should be examined in detail. The second relates to Se biofortification and phytoremediation. Phytoremediation of Se is popular in Se-rich places. The plants harvested in phytoremediation could be used as the organic source of Se. However, attention must be paid to the fact that there are possibly other heavy metals in the harvested plants. In addition, transgenic plant technology has been used for phytoremediation and Se biofortification. Since some people are very sensitive to genetically modification of crop plants, application of transgenic plants for Se biofortification should be careful. The third relates to theuse of Se hyperaccumulators. Some Se hyperaccumulators, such as *Stanleya pinnata* and *Astragalus bisulcatus* and *Cardamine enshiensis*, should be paid more attentions, especially *C. enshiensis*, because it is edible and can be directly used in food. The forth relates to the use of Se nanoparticles. Relatively, Se nanoparticles are less toxic and more eco-friendly for both humans and the environment. More researches are necessary for use of Se nanoparticles, especially for the production of Se nanoparticles using plants and fungi. The fifth relates to increased plant resistance to abiotic stresses. Exogenous Se compounds and seleno-bacteria synergistically improve plant resistance to abiotic stresses, thus the synergism should be well used for plant resistance to abiotic stresses, especially drought and salt stress. Finally, the sixth relates to functions of root endophytic fungi and dark septate fungi. The two types of fungi possess ecological functions similar to mycorrhizal fungi, and they often colonize many plant species. However, very little attention has been paid to them.

## Figures and Tables

**Table 1 jof-06-00059-t001:** Arbuscular mycorrhizal fungi (AMFs) and root endophytic fungi (REFs) often used for Se biofortification.

Microbes	Microbial Types	Host Plants	References
*Funneliformis mosseae*	AMF	*Triticum aestivum*, *Lactuca sativa*, *Asparagus officinalis*,	[105,106,107,108]
*Glomus claroideum*	AMF	*Triticum aestivum*	[109]
*Glomus fasciculatum*	AMF	*Allium sativum*	[110]
*Glomus irtraradices*	AMF	*Allium sativum*	[55]
*Glomus mosseae*	AMF	*Lolium perenne, Allium sativum, Medicago sativa, Glycine max, Zea mays*	[110,111,112]
*Glomus versiform*	AMF	*Triticum aestivum*	[105]
*Rhizophagus intraradices*	AMF	*Lactuca sativa, Asparagus officinalis, Lactuca sativa, Allium cepa*	[106,107,108,113]
*Alternaria seleniiphila*	REF	*Stanleya pinnata*	[114]
*Alternaria astragali*	REF	*Astragalus bisulcatus*	[114]
*Aspergillus leporis*	REF	*Stanleya pinnata*	[114]
*Fusarium acuminatum*	REF	*Astragalus racemosus*	[114]
*Trichoderma harzianum*	REF	*Allium cepa*	[106]

**Table 2 jof-06-00059-t002:** Plant growth-promoting rhizobacteria (PGPRs) often used for Se biofortification.

Microbes	Host Plants	References
*Acinetobacters* sp. *E6.2*	*-*	[132]
*Acinetobater* sp.	*Triticum aestivum*	[139]
*Alcaligenes faecalis*	*Ricinus communis, Glycine max*	[35]
*Anabaena* sp.	*Triticum aestivum*	[30,140]
*Bacillus amyloliquefaciens*	*Arabidopsis thaliana*	[141]
*Bacillus axarquiens*	*Triticum aestivum*	[139]
*Bacillus cereus*	*Triticum aestivum*	[34]
*Bacillus licheniformis*	*Triticum aestivum*	[34]
*Bacillus mycoides*	*Brassica juncea*	[142]
*Bacillus pichinotyi*	*Triticum aestivum*	[33]
*Bacillus* sp. *E5*	*-*	[132]
*Bacillus* sp. *E6.1*	*Triticum aestivum*	[139]
*Bacillus* sp. *R12*	*Triticum aestivum*	[109]
*Bacillus subtilis*	*Allium cepa*	[113]
*Calothrix* sp.	*Triticum aestivum*	[30,140]
*Enterobacter ludwigii*	*Triticum aestivum*	[139]
*Enterobacter* sp. *B16*	*Triticum aestivum*	[109]
*Klebsiella oxytoca*	*Triticum aestivum*	[139]
*Paraburkholderia megapolitana*	*Ricinus communis, Glycine max*	[35]
*Providencia* sp.	*Triticum aestivum*	[30,140]
*Pseudomnas* sp. *R8*	*Triticum aestivum*	[109,133]
*Rhizobium* sp.	*Astragalus bisulcatus, A. drummondii*	[138]
*Rhizosphere bacteria*	*Scirpus robustus, Polypogon monspeliensis*	[143]
Se-tolerant bacteria	*Brassica juncea*	[144]
*Stenotrophomonas maltophilia*	*Ricinus communis, Glycine max, Brassica juncea*	[35,142]
*Stenotrophomonas* sp. *B19*	*Triticum aestivum*	[109,133]

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
