# Peer review of "Selenium Biofortification of Crop Food by Beneficial Microorganisms"

_jof, 2020, doi:10.3390/jof6020059_

Round 1

Reviewer 1 Report

This manuscript summarizes news regarding selenium biofortification of crop food by beneficial microorganisms. The subject of this manuscript is consistent with the scope of this Journal. These results are interesting and very important for future of agricultural biotechnological. Manuscript is almost well written, this work have adequate to many references, showing a good understanding of the topic. However, the conclusions not exactly corresponds with the work's content.

Manuscript can be published in scientific Journal of Fungi after some changes (major revision):

  • In my opinion that manuscript is missing table and image. This manuscript is dry presentation of literature. Please suggest these parts.
  • I think that the chapter concluding remarks and perspective should not contain a citations. This is a critical chapter of the authors based on known literature.Please rebuild this so that order prevails.
  • Please check the names of bacteria and fungi carefully, adjusting them to taxonomy requirements.
  • Please, be sure that all the references cited in the manuscript are also included in the reference list and vice versa with matching spellings and dates.
  • In my opinion, manuscript needs a major changes to be published. Manuscript contains few aspects of criticized authors.

Author Response

Dear Editors

We revised the text of the first manuscript.

About some questions from the two reviewers, we made answers listed below:

Answers to Reviewer 1

(1) we revised a little the conclusion.

(2) a table was added, and some BMOs and their host plants were listed in the table.

(3) in the part “Conclusive remarks and perspectives”, some references were cited. We think it is not impossible, because some review articles used references in the part. For example:

Domka AM, Rozpa ˛dek P and Turnau K (2019) Are Fungal Endophytes Merely Mycorrhizal Copycats? The Role of Fungal Endophytes in the Adaptation of Plants to Metal Toxicity. Front. Microbiol. 10:371. doi: 10.3389/fmicb.2019.00371

Some references are cited in order to elucidate/explain viewpoints.

(4) we checked all the Latin names of the bacteria and fungi.

(5) the software Endnote was used to manage all the references, thus all the references cited in the article were listed in the bibliography. The references not cited in the article have been deleted.

Best regards!

Wu Chu

Reviewer 2 Report

The research reported in this manuscript is good quality, however, it requires better framing (explain what is known and what is not necessarily known, justifying publication in the international journal -selenium biofortification of crop food by beneficial microorganisms is probably the least known). On the contrary, no apparent serious flaws are there either in the analytics or the stats, but some points need a better explanation, probably. I have listed specific comments which may guide the authors through revisions.

I think the focus of the paper makes it of interest to JoF readership. The topic in this study is of interest to the readers and expands our knowledge on the effect of PGPR for Se uptake in crop foods. The present version of the manuscript has not many apparent inconsistencies and just needs a specific revision.

The structure of the article conform to an acceptable format, it would only be necessary to put the acknowledgments section before references for the manuscript to comply with the required format. However, the reading and understanding of the manuscript are difficult to follow. There is an improper use of grammar, tense and spelling throughout the manuscript, including the titles and legends of tables and figures, which make it confusing and ambiguous which affects precision and clarity of the writing. Besides, the manuscript is mainly written in first person language instead of third-person language, which affects the formality and scientific style of the writing.

The role of AMF in plant Se uptake is addressed in the introduction, however, it is necessary to add further evidence that supports the importance of the research work.

Comment

Lines 95-96: Please add references to improve your paper for making readers given more information about the recent AMF studies as follows:

Smith, S.E.; Read, D.J. Arbuscular mycorrhizaes. In Smith, S.E.; Read, D.J. (Eds), Mycorrhizal symbiosis 3rd Edition. Academic Press: London, UK, 2008; pp.13–187.

Smith, S.E.; Jakobsen, I.; Grønlund, M.; Smith, F.A. Roles of arbuscular mycorrhizas in plant phosphorus nutrition: interactions between pathways of phosphorus uptake in arbuscular mycorrhizal roots have important implications for understanding and manipulating plant phosphorus acquisition. Plant Physiol. 2011, 156, 1050–1057.

Higo, M.; Takahashi, Y.; Gunji, K.; Isobe K. How are arbuscular mycorrhizal associations related to maize growth performance during short-term cover crop rotation? J. Sci. Food Agric. 2018, 98, 1388–1396.
Higo, M.; Tatewaki, Y.; Gunji, K.; Kaseda, A.; Isobe, K. Cover cropping can be a stronger determinant than host crop identity for arbuscular mycorrhizal fungal communities colonizing maize and soybean. Peer J. 2019, 7, e6403.
Higo, M.; Tatewaki, Y.; Iida, K.; Yokota, K.; Isobe, K. Amplicon sequencing analysis of arbuscular mycorrhizal fungal communities colonizing maize roots in different cover cropping and tillage systems. Sci. Rep. 2020, 10, 6093.

Comment

Lines 291-292: “But it must be paid on that there are possibly other heavy metals in the harvested plants.” Please consider adding the reference to improve your paper and provide information you meant for related readers as follows:

Higo, M.; Kang, D.J.; Isobe, K.First report of community dynamics of arbuscular mycorrhizal fungi in radiocesium degradation lands after the Fukushima-Daiichi Nuclear disaster in Japan. Sci. Rep. 2019, 9, 8240.

This review article is not enough to clear and it is necessary to show more evidence to fully support the proposed Se uptake by beneficial microorganisms. Even though the review is clearly stated in the manuscript, it seems a little bit ambiguous as it states that “Beneficial organisms inoculation improve Se uptake of the crop foods”.

In my opinion, the article is interesting and presents valuable information about beneficial microorganisms and Se uptake. Overall, I strongly suggest revising the text for English, and the clarity of exposition. It is also recommended to thoroughly review the manuscript and to consult with a professional translator or with someone with the relevant expertise to give guidance on English writing. Please check the flow and structures by a native English speaker to better understanding the paper. Unfortunately, English and clarity of the manuscript needed to be improved so much, and the text sounds quite unprofessional.

Author Response

Dear Editors

We revised the text of the first manuscript.

About some questions from the two reviewers, we made answers listed below:

Answers to Reviewer 2

(1) About the roles of AMFs in Se biofortificaiton, we revised the text, and made it clear. Because ectomycorrhizal fungi colonize in roots of gymnosperms, they are not used for food crops.

(2) In the manuscript, the last paragraph was written in the first person language. In other parts, the manuscript was written using passive voice.

(3) About references supplement in Lines 95-96, we have done the references provided by the reviewer.

(4) About references supplement in Lines 291-292, we have done the reference provided by the reviewer.

(5). All the whole text has been revised and improved the language.

Best regards!

Wu Chu

Round 2

Reviewer 1 Report

Manuscript can be published in scientific Journal of Fungi after some changes (minor revision):

  • In my opinion, the placed table should be divided into two tables. In table 1, please organize fungal data (chapter 2.1), table 2 should contain bacterial data - chapter 2.2.
  • “I think the chapter concluding remarks and perspective should not contain a citation. This is a critical chapter of the authors based on known literature. Please rebuild this so that order prevails” - I think this chapter has not been improved enough.
  • Please check the names of bacteria and fungi carefully, adjusting them to taxonomy requirements – for example name of strain should be written normal style.

I kindly ask the authors to improve the manuscript. Then I can recommend manuscript for publication.

Author Response

Dear Editors

According to the reviewer’s opinions, we revised the manuscript. We made answers to these opinions below.

(1) The table was divided into two tables according to types of beneficial microorganisms.

(2) In the part “Concluding remarks and perspective”, some references were deleted, and some sentences were revised.

(3) All the Latin names of microorganisms were checked out.

(4) Some spelling errors were revised in all the text, shown in blue. Some revised places were shown in blue. In addition, few sentences were deleted.

Best regards!

Wu Chu